# Chemical Cues Used by the Weevil *Curculio chinensis* in Attacking the Host Oil Plant *Camellia oleifera*

Hualong Qiu [1], Danyang Zhao [1], Eduardo G. P. Fox [2], Siquan Ling [1], Changsheng Qin [1] and Jinzhu Xu [1,*]

1 Guangdong Provincial Key Laboratory of Silviculture, Protection and Utilization, Guangdong Academy of Forestry, Guangzhou 510520, China
2 Programa de Pós-Graduação em Ambiente e Sociedade (PPGAS), Universidade Estadual de Goiás (UEG), Jardim América, Quirinópolis 75650-000, Brazil
* Correspondence: xujinzhu@sinogaf.cn

**Abstract:** The weevil *Curculio chinensis* Chevrolat (Coleoptera: Curculionidae) is a major cause of economic losses to growers of *Camellia oleifera* in China, as females lay their eggs in developing fruits and the hatching larvae feed on their seed, aborting fruit growth. Olfactory cues play a key role in the host location of this weevil. The present study focused on identifying volatiles from different parts of the host plant *Ca. oleifera,* namely, the leaves, fruit peel, and seeds, and testing the antennal and behavioral responses of adult *Cu. chinensis* to those same volatiles. Methods relied on gas chromatography, electroantennograms, and Y-tube bioassays. The results included a total of twenty-five volatiles emitted by the three plant parts, among which eight elicited antennal responses in *Cu. chinensis* adults of both sexes. The behavioral bioassays indicated that 3-hexenal, trans-2-hexen-1-ol, methyl salicylate, geraniol, and phenethyl alcohol were attractive to *Cu. chinensis,* while trans-2-hexenal and 2-ethyl-1-hexanol were repellent. Tests with different concentrations indicated that the behavioral response could be dose-dependent. Future studies should focus on field tests with blends of the attractant compounds in order to develop novel, improved control methods for field applications.

**Keywords:** integrated pest management; plant odors; secondary metabolites; green chemistry

## 1. Introduction

*Camellia oleifera* is one of the four major commercial woody oil plants in the world, displaying a wide range of applications in forestry, agriculture, fishery, and in the food and chemical industries [1,2]. The fragrant oil extracted from *Ca. oleifera* seeds is edible and regarded as healthy and nutritious, while also devoid of substances harmful to the human body, such as erucic acid, cholesterol, and aflatoxin [3]. In fact, *Ca. oleifera* seed oil has a number of benefits for human health, for instance, as a cosmeceutical that protects against ultraviolet radiation, delays skin aging, and promotes wound healing, among other benefits [1]. As a result, *Ca. oleifera* seed oil is praised as "the best edible oil in China". The rapid development of advanced processing plants of tea oil technology in China was accompanied by the consolidation of a complete production chain, from harvesting and production down to deep oil processing [4]. As the cultivated area of *Ca. oleifera* forests continues to expand, the frequency and scale of the harm caused by diseases and pests of *Ca. oleifera* have intensified, seriously impacting the gross crop yield and final product quality, and imposing a major threat to the sustained development of the *Ca. oleifera* industry [5].

The weevil *Curculio chinensis* (Coleoptera, Curculionidae) is one of the most damaging pests, causing a low yield in *Ca. oleifera* production [6,7]. It is widely distributed throughout the major land areas in China producing *Ca. oleifera*, chiefly in the provinces of Yunnan, Guizhou, Guangxi, Zhejiang, Hunan, Jiangxi, and Guangdong [8]. Adult *Cu. chinensis* typically emerge from the soil during late April to May, and feed on young fruits for supplemental nutrition. After successful mating, female beetles lay eggs in the seeds of



Camellia trees and the newly hatched larvae feed on seeds before leaving the host plant. Under normal conditions, larvae emerge from the seeds during early July to late September and overwinter in the soil, where they pupate and emerge as adults after 2 years [9,10]. Due to the fact that *Cu. chinensis* larvae hide inside fruit, it is difficult to monitor and forecast the appearance of adults for timing applications, which affect mating or target freshly emerged larvae [11]. Therefore, developing efficient methods for the monitoring and management of *Cu. chinensis* adults has become an urgent priority.

Insects mainly locate their host plants and select for appropriate oviposition sites based on received recognizable plant volatiles using various types of chemosensory sensilla on their antennae [12–15]. In their exploitation of host plants, weevils mainly utilize aggregation pheromones and plant odors to search for optimal mating, oviposition, and feeding sites (Figure 1) [16–18]. The chemical cues used by insects to locate host plants are also employed in forecasting and controlling agricultural and forestry pests; however, currently, the chemicals exploited by *Cu. chinensis* with *Ca. oleifera* have not yet been described. Therefore, the present study focused on the electrophysiological and behavioral responses of *Cu. chinensis* adults to the volatiles emitted from the leaves, fruit peels, and seeds of *Ca. oleifera*. The obtained results revealed potential attractants for trapping *Cu. chinensis*.

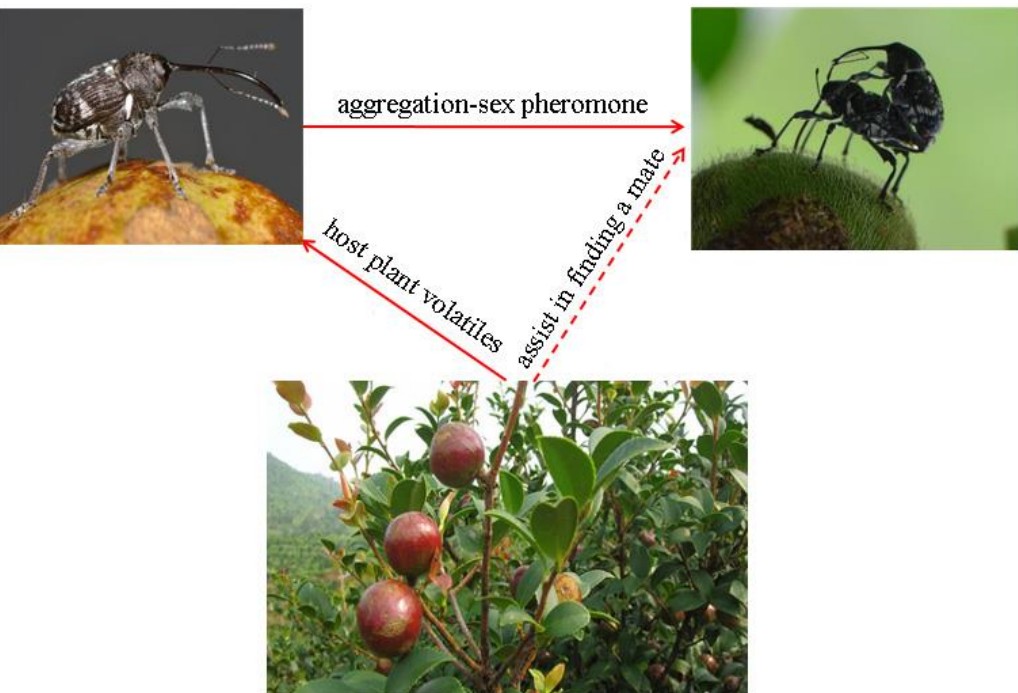

**Figure 1.** The schematic diagram of adult *Curculio chinensis* beetles locating a mate and feeding on fruit using host plant odors and their aggregation pheromone.

## 2. Materials and Methods

### 2.1. Insect Collection

Adults of *Cu. chinensis* were collected from the Xiaokeng and Longshan forestry stations in Shaoguan City, Guangdong province, from April to August of the years 2018 and 2019. Adults of *Cu. chinensis* were captured by shaking their host trees, from which they typically fell while feigning death (thanatosis behavior) into a white fishing spinning net placed under the tree. From this net, adults were manually relocated into insect-rearing cages (length × width × height = 20 cm × 15 cm × 9 cm) with a lid. Laboratory rearing conditions for *Cu. chinensis* were 25 ± 1 °C, with a relative humidity of approximately 65 ± 5%, under a photoperiod L:D = 12 h:12 h. Washed fresh *Ca. oleifera* fruits were provided to the *Cu. chinensis* daily.

### 2.2. Extraction and GC-MS Identification of Host Plant Chemicals

Chemicals from the leaves, fruit peel, and seeds of *Ca. oleifera* were extracted with a solvent and identified with mass spectrometry. The leaves were green leaves that had just sprouted, the length of which was approximately 3–5 cm and the width was approximately 2–3 cm. The fruits were immature and green, with a diameter of 2–3 cm, and the length of seeds was 0.8–1.2 cm. Specifically, approximately 5 g of each whole plant part per sample was placed in a beaker, to which *n*-hexane (Sigma-Aldrich, Burlington, MA, USA, ≥99% purity) was added in enough quantity (50–100 mL) to submerge the sample. The beaker was then wrapped with tin foil and sealing film and placed on a shaking table at a speed of 130 r/min at 25 °C for 1 h. Anhydrous sodium sulfate was then added to the remaining solvent mixture in a beaker and gently stirred for 2 min, to remove excess water. Each sample solution was then filtered through an organic filter membrane of 0.45 μm pore size, and, finally, the solution was evaporated to 0.5 mL under a nitrogen flow. The concentrated samples were stored in a −20 °C freezer until analysis.

Compounds were identified with an Agilent 5977B mass selective detector (MSD) coupled with an 8890 GC system (Agilent, Santa Clara, CA, USA). The GC system was equipped with a HP-5MS capillary column (30 m × 25 μm × 0.25 μm). The oven temperature started at 40 °C held for 1 min, raising at a rate of 10 °C/min to 220 °C, and maintained at 220 °C for 1 min. Helium was the carrier gas at a flow rate of 1 mL/min. The injector temperature was 220 °C and the ion source temperature was 230 °C. Compounds were identified through a comparison with external standards and the internal library NIST17. Commercial specifications of standard compounds are given in Table S1 of the Supplementary Files.

### 2.3. GC-EAD of Cu. Chinensis Volatiles

In order to screen for potential attractants to *Cu. Chinensis*, the antennal electrophysiological responses of adult *Cu. Chinensis* females and males were tested with GC-EAD to the volatiles of *Ca. oleifera* leaves, fruit peel, and seeds. The GC-EAD system was equipped with an Agilent 7890b GC with a flame ionization detector (FID) and an electroantennographic detector (EAD, Syntech, Hilversum, Netherlands). The GC system was set as described above for the GC-MS. The temperature at the FID detector was 260 °C.

The antennae of *Cu. Chinensis* were excised at the base, and a slight opening 0.5 mm wide was cut at each tip. Each end of the excised antennae was then connected to two glass capillary electrodes of EAG. A 0.9% NaCl solution was poured into each glass capillary. The rate of constant airflow that passed through the insect antennae was 2 L/min. The external EAD signal was amplified 10 times; there was no external amplifier for the FID signal. The sample injection volume was 1 μL, injected at the splitless injection mode. The proportion of samples entering the column and the antenna was 1:1. Six antennae from different individuals (males and females) were tested as replicates for each sex at each tested sample concentration.

### 2.4. Behavioral Bioassays to Single Compound with Y-Tube Olfactometer

The behavioral responses of the *Cu. chinensis* adults to the identified compounds eliciting a GC-EAD response described in Section 2.3 were observed using a Y-tube olfactometer. The experimental conditions during the *Cu. chinensis* behavioral assays were maintained at 25 ± 1 °C and a relative humidity of 65 ± 5%. The internal diameter in the Y-tube was 2.0 cm, the length of each arm was 25 cm, and the length of the middle main tube was 20 cm; the angle between the arms was 75°. In total, 5 uL of each different concentration of standard compounds was applied onto 1 cm sided squares of filter paper, allocated in an inserted 200 mL glass bottle as the source odor. Another filter paper square containing 5 uL of *n*-hexane was used as the negative control. The standard compounds were 3-hexenal, *trans*-2-hexenal, *trans*-2-hexen-1-ol, 2-ethyl-1-hexanol, phenethyl alcohol, methyl salicylate, geraniol, and eugenol (Table S1). The air flow from the odor source and control bottles entered the system through activated carbon and was humidified with

twice-distilled deionized water. The air flow speed in each arm was set to 150 mL/min. Behavioral assays were undertaken in a darkened room, with a 30 W incandescent lamp placed on top of the Y tube so as to ensure that the light intensity was evenly distributed across the Y-tube olfactometer. All behavioral tests took place between 8:00 and 12:00 a.m., as, based on personal investigation, this is when *Cu. chinensis* are typically more active. Solutions of each compound at 100, 10, and 1 μg/mL were prepared for testing.

One adult *Cu. chinensis* was introduced into the base of the Y-tube. A given sample was interpreted to induce a positive attraction response whenever the beetle went into one arm beyond 2/3 of its length and stayed in that area of the arm for longer than 0.5 min. If a beetle did not make a choice, which meant it stayed in the middle main tube for 5 min, it was recorded as showing 'no response'. After every insect choice, the odor source was refreshed. Between 50 and 60 *Cu. chinensis* adults were tested for each sample concentration, and each beetle was tested only once for each sample concentration. All the tested individuals were starved for 24 h before the behavioral tests. In order to eliminate a possible bias of a positional effect on the behavioral results, the Y-tube arms were alternated between trials; also, the inner wall of the Y-tube was cleaned with 75% ethanol and dried with a blower every 3 trials. After each treatment test, the glass bottle and Y-tube were immersed in 50% ethanol solution for ultrasonic cleaning for 10 min to eliminate any lingering odors between trials.

*2.5. Statistical Analyses*

For the preliminary evaluation of the GC-MS data and finding out the outliers, a principal component analysis (PCA) was performed. In order to find out the key differences in volatiles between the leaves, fruit peel, and seeds, biomarkers were detected based on orthogonal partial least squares discriminant analysis (OPLS-DA) regression coefficients ($p < 0.05$), and standard errors were calculated using jack knifing at 95% confidence intervals. For the Y-tube bioassays, the null hypothesis was that insects showed no preference for either arm (i.e., 50:50% response), and was tested using chi-square goodness-of-fit tests with SPSS software v.20. A few individuals (roughly 5% of tested insects) that did not make a choice were excluded from the analysis.

**3. Results**

*3.1. Identification of Plant Volatiles with GC-MS*

A total of 25 volatiles was identified from different plant parts of *Ca. oleifera* using GC-MS spectra. Specifically, 16 volatiles were identified from leaves, 16 from fruit peels, and 15 volatiles from seeds (Figure 2a and Table 1). The total ion flow chromatograms of leaves, fruit peels, and seeds, as well as the heatmap built on the relative abundance of each compound, are given in Figure S1 of the Supplementary File. Eight such volatiles were simultaneously detected from all three plant parts, where three volatiles (namely, 2,6,10-trimethyl tetradecane, 2-methyl eicosane, and *n*-eicosane) were detected exclusively from the leaves, and α-pinene and β-caryophyllene exclusively from fruit peels. Of special note, six chemicals (fumaric acid, linalool, phenethyl alcohol, methyl salicylate, eugenol, and α-farnesene) were exclusive to seeds. Leaves and fruit peels from the analyzed species shared five chemicals, α-cedrene, butylated hydroxytoluene, 2,6,11,15-tetramethyl-hexadecane, *n*-octadecane, and 2,6,10,15-tetramethyl-heptadecane, none of which were present in seeds. Geraniol was the sole volatile shared between fruit peels and seeds, but not detected in the leaves. Clearly, seeds presented the most differentiated chemical profile.

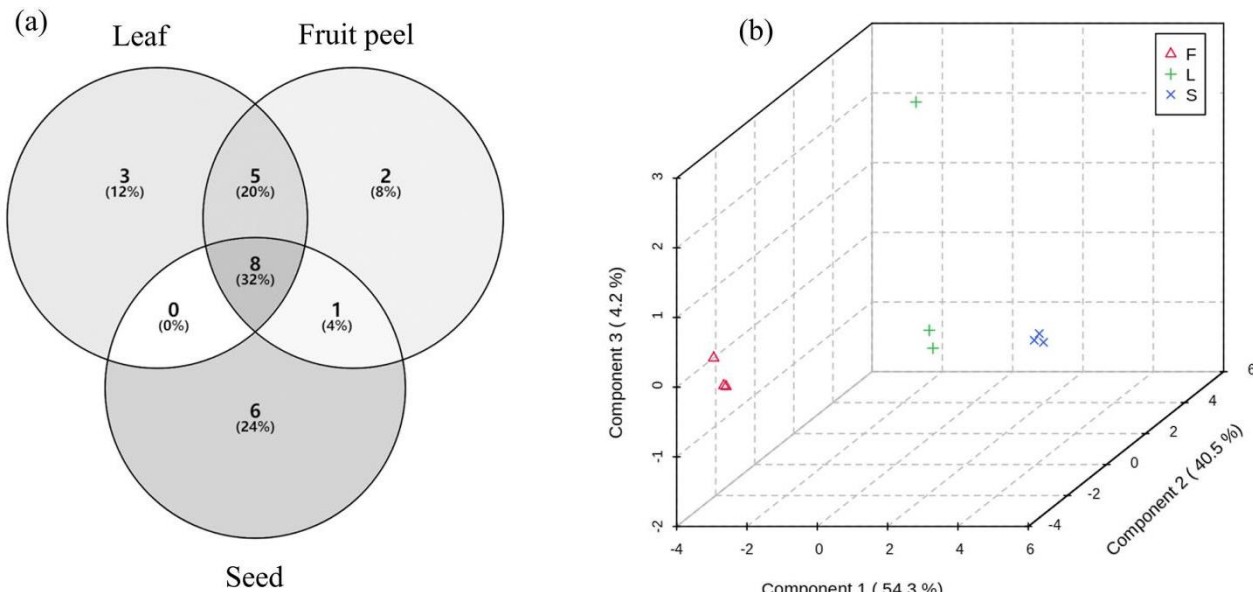

**Figure 2.** Venn diagram of volatile numbers in different plant tissues (**a**) and 3D score plots between the selected components (**b**). Letters F, L, and S in the legend of subpart (**b**) stand for fruit peel, leaves, and seeds, respectively.

**Table 1.** Volatile chemicals and their relative proportions as emissions from *Camellia oleifera* leaves, fruit peels, and seeds (N = 3).

| No. | Chemical Name | RT (min) | Relative Proportion % (Mean ± SE) | | |
|---|---|---|---|---|---|
| | | | **Leaves** | **Fruit Peel** | **Seeds** |
| 1 | 3-Hexenal * | 3.34 | 18.99 ± 1.06 | 2.70 ± 0.15 | 4.50 ± 0.22 |
| 2 | *trans*-2-hexenal * | 4.04 | 10.09 ± 0.54 | 15.62 ± 0.58 | 13.18 ± 1.03 |
| 3 | *trans*-2-Hexen-1-ol * | 4.27 | 3.90 ± 0.32 | 5.93 ± 1.64 | 3.77 ± 0.06 |
| 4 | α-Pinene | 5.13 | / | 4.90 ± 0.25 | / |
| 5 | Fumaric acid | 6.35 | 3.46 ± 0.78 | 5.50 ± 0.15 | 1.42 ± 0.67 |
| 6 | 2-Ethyl-1-hexanol * | 6.57 | / | / | 3.44 ± 0.11 |
| 7 | γ-Terpinene | 7.08 | 3.83 ± 0.25 | 9.01 ± 0.23 | 3.13 ± 0.22 |
| 8 | Linalool | 7.65 | / | / | 1.98 ± 0.08 |
| 9 | Phenethyl alcohol * | 7.88 | / | / | 10.09 ± 0.11 |
| 10 | Terpinen-4-ol | 8.90 | 2.94 ± 0.18 | 5.34 ± 0.12 | 1.66 ± 0.92 |
| 11 | Methyl salicylate * | 9.14 | / | / | 2.49 ± 0.13 |
| 12 | Geraniol * | 9.92 | / | 2.39 ± 0.30 | 27.64 ± 0.74 |
| 13 | Eugenol * | 11.40 | / | / | 17.94 ± 0.36 |
| 14 | *n*-Tetradecane | 11.89 | 11.65 ± 0.76 | 7.97 ± 0.70 | 2.23 ± 0.14 |
| 15 | α-Cedrene | 12.25 | 2.60 ± 0.23 | 3.49 ± 0.82 | / |
| 16 | β-Caryophyllene | 12.33 | / | 9.16 ± 0.13 | / |
| 17 | Tetradecane, 2,6,10-trimethyl- | 12.69 | 2.67 ± 0.07 | / | / |
| 18 | α-Farnesene | 13.30 | / | / | 3.93 ± 0.52 |
| 19 | Butylated hydroxytoluene | 13.41 | 2.51 ± 0.27 | 2.79 ± 0.16 | / |
| 20 | *n*-Hexadecane | 14.35 | 12.26 ± 1.23 | 12.32 ± 0.87 | 2.60 ± 0.09 |
| 21 | Hexadecane, 2,6,11,15-tetramethyl- | 15.63 | 4.87 ± 0.17 | 3.67 ± 0.09 | / |
| 22 | *n*-Octadecane | 16.57 | 10.09 ± 0.55 | 5.79 ± 0.13 | / |
| 23 | Heptadecane, 2,6,10,15-tetramethyl- | 17.88 | 4.78 ± 0.22 | 3.41 ± 0.07 | / |
| 24 | Eicosane, 2-methyl- | 18.31 | 2.80 ± 0.11 | / | / |
| 25 | *n*-Eicosane | 18.59 | 3.26 ± 0.14 | / | / |

Note: An asterisk indicates compounds eliciting antennal EAD activity in the Camellia weevil, *Curculio chinensis*.

The principal components analysis (PCA) revealed clear clusters with no outliers, which were so closely related ($p < 0.05$) that the resolution was limited (data not shown). On the other hand, OPLS-DA enabled a clear discrimination between clusters of volatiles

from different plant parts. This cluster analysis returned high explanation coefficients ($R^2X = 0.67$ and $R^2Y = 0.93$), and a predictive ability of $Q^2_{(cum)} = 0.89$ (with two principal components) (Figure 2b), providing marked sample clustering, illustrating the unique chemistry of bouquets from leaves, fruit peels, and seeds.

### 3.2. GC-EAD Analysis

The GC-EAD results for adult *Cu. chinensis* in Figure 3 show an antennal response to two of the leaf volatiles. Their retention times were 3.34 min and 4.04 min, respectively, identified as 3-hexenal and *trans*-2-hexenal, based on external injections with standard chemicals. Fruit peels produced three volatiles inducing antennal responses in *Cu. chinensis*, two of which were the same as that just mentioned for the leaves. The third volatile had a retention time of 4.27 min, likewise identified as *trans*-2-hexen-1-ol by comparison with chemical standards. Most remarkably, eight chemicals from fruit seeds induced antennal responses, of which just two were the same as with the other plant parts (3-hexenal and *trans*-2-hexenal), another two (*trans*-2-hexen-1-ol and geraniol) were shared only with peels, and the remaining four were unique to seeds (2-ethyl-1-hexanol, phenethyl alcohol, methyl salicylate, and eugenol).

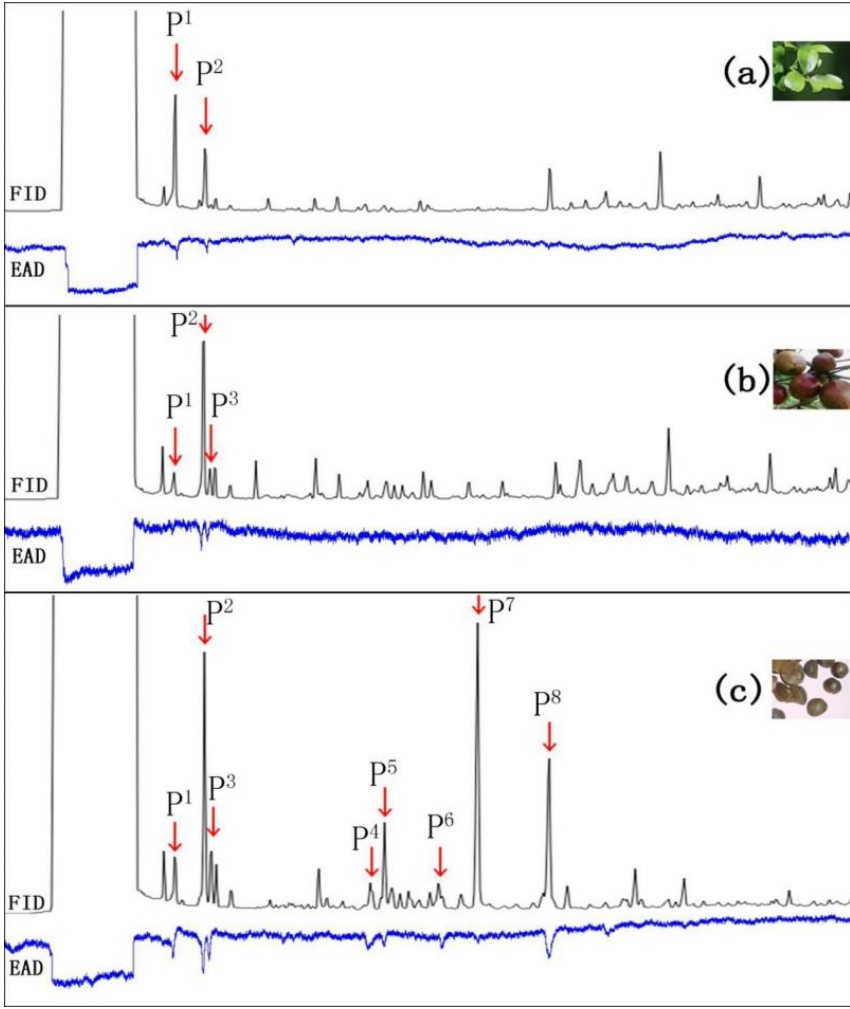

**Figure 3.** GC-EAD analysis of volatiles from *Camellia oleifera* leaves (**a**), fruits (**b**), and seeds (**c**) to the antennae of *Curculio chinensis*, indicating that a total of eight peaks was electrophysiologically active compounds. P1-8 were 3-hexenal, *trans*-2-hexenal, *trans*-2-hexen-1-ol, 2-ethyl-1-hexanol, phenethyl alcohol, methyl salicylate, geraniol, and eugenol, respectively.

*3.3. Behavioral Responses in Cu. chinensis to the GC-EAD Active Compounds*

As demonstrated by the Y-tube olfactometer bioassay results in Figure 4, *Cu. chinensis* adults responded with variable behaviors to the GC-EAD active compounds, suggesting a concentration-dependent response intensity. Specifically, 100 µg/mL of 3-hexenal exerted an attraction response ($\chi^2 = 4.421$, df = 1, and $p = 0.036$), while *trans*-2-hexenal was repellent at 100 µg/mL and at 10 µg/mL (respectively, $\chi^2 = 37.79$, df = 1, $p < 0.001$; $\chi^2 = 9.6$, df = 1, and $p = 0.002$). The repellent effect decreased at the lower concentration of 1 µg/mL ($p > 0.05$). *Trans*-2-hexen-1-ol at the highest concentration of 100 µg/mL was highly attractive to *Cu. chinensis* ($\chi^2 = 11.07$, df = 1, $p = 0.001$), but the effect decreased at 10 µg/mL ($\chi2 = 3.63$, df = 1, $p = 0.057$), ultimately disappearing at the concentration of 1 µg/mL ($p > 0.05$). Additionally, 2-ethyl-1-hexanol elicited significant repellency at 100 µg/mL and 10 µg/mL (100 µg/mL: $\chi^2 = 32.44$, df = 1, and $p < 0.001$; 10 µg/mL: $\chi^2 = 9.28$, df = 1, and $p = 0.002$), and the effect decreased significantly—but remained—at a concentration of 1 µg/mL ($p > 0.05$). *Cu. chinensis* showed attraction to the 100 µg/mL of phenethyl alcohol ($\chi^2 = 9.31$, df =1, $p = 0.021$), and the effect decreased significantly at lower concentrations ($p > 0.05$). Methyl salicylate was only attractive to *Cu. chinensis* at concentrations of 100 µg/mL and 10 µg/mL (100 µg/mL: $\chi^2 = 11.25$, df = 1, and $p = 0.001$; 10 µg/mL: $\chi^2 = 5.33$, df = 1, and $p = 0.021$). Similarly, geraniol was attractive to *Cu. chinensis* at the highest and intermediate concentrations, but not at the lowest (100 µg/mL: $\chi^2 = 13.75$, df = 1, and $p < 0.001$; 10 µg/mL: $p = 0.013$, and $\chi^2 = 6.23$). Eugenol was remarkable in generating no significant attractive nor repellent effects ($p > 0.05$), in spite of producing a strong GC-EAD response.

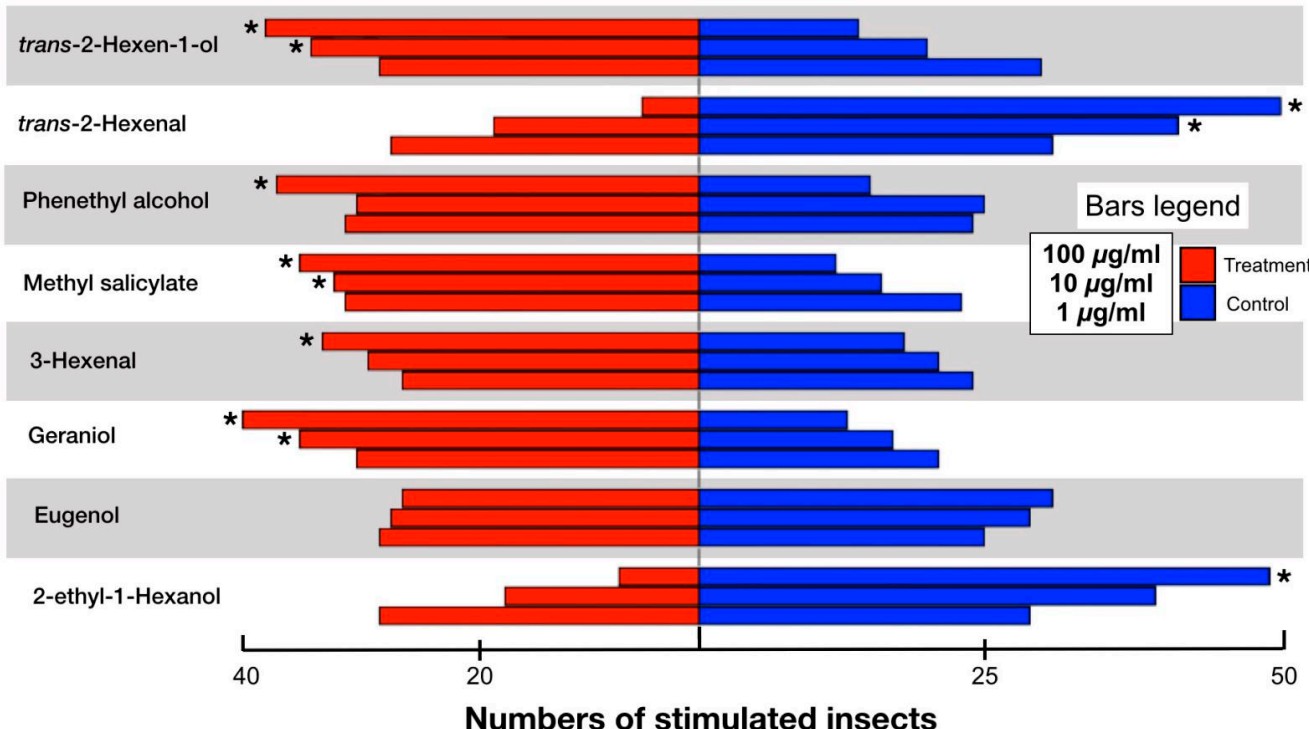

**Figure 4.** Behavioral responses of the weevil *Curculio chinensis* to different concentrations of electrophysiologically active volatiles identified from the leaves, fruits, and seeds of *Camellia oleifera*. Sets of bars indicate the different concentrations of 100, 10, and 1 µg/mL (from top down); asterisks indicate statistical significance from the control group ($p < 0.05$).

## 4. Discussion

Plant volatiles are important cues used by pest insects in locating suitable hosts [19,20]. This study provided the first record of host plant volatiles generating behavioral and electrophysiological responses by *Cu. chinensis*, focusing on different parts of *Ca. oleifera*.

We chose the three plant parts—leaves, fruit peels, and seeds—excluding flowers, as the active season of *Cu. chinensis* adults coincides with leaf buds and young fruits developing in the host plants [11]. Thereafter, the adult beetles find and feed from fresh leaves and fruit, causing damage. Consequently, it is reasonable to expect that the volatiles from *Ca. oleifera* leaves and fruit should play a central role in host location for *Cu. chinensis*.

The volatiles of the different host plant parts and their blends varied significantly. Eight volatiles were common to all plant parts, and five were shared by the leaves and fruit peels. Volatiles from such exposed plant parts are most likely involved in long-distance detection by pest insects [12,14]. Some volatiles were exclusive to certain plant parts, only three in leaves, two exclusive to fruit peels, and six volatiles were exclusively found in seeds. The fact that seeds have a rich, unique mixture of odors was, thus, confirmed. We hypothesized that exclusive volatiles serve as short-distance cues towards specific plant organs [21]. As such, such region-specific volatiles would provide key information directing to the most appropriate feeding and oviposition sites for *Cu. chinensis*.

GC-EAD is an analytical method allowing for the rapid screening of volatiles stimulating the olfactory sensilla of insects out of complex chemical mixtures [22]. Therefore, a GC-EAD analysis can be useful for identifying potentially useful compounds affecting insect behavior. Adult *Cu. chinensis* yielded EAG responses to a total of eight plant volatiles, from all of the different plant parts. These bioactive volatiles were expected to somehow influence the behavior of *Cu. chinensis*.

The bioactive compounds were selected to be tested at different concentrations for selective behavior of *Cu. chinensis* in a Y-tube olfactometer. Among these, five compounds worked as attractants to *Cu. chinensis*: 3-hexenal, *trans*-2-hexen-1-ol, phenethyl alcohol, methyl salicylate, and geraniol. Visser [23] observed that the EAG antennal responses in the Colorado beetle *Leptinotarsa decemlineata* relied on strong chemical selectivity by their antennal receptors, where some green leaf volatiles, such as *trans*-2-hexen-1-ol and methyl salicylate, could elicit a strong EAG response even at quite low concentrations. In this study, the compound 3-hexanal was attractive to *Cu. Chinensis;* the same chemical proved to be an attractant to the pest moth *Ectropis oblique* [24]. Therefore, the five responsive volatiles were likely key for host recognition and location by *Cu. chinensis* in the field.

Only two compounds, *trans*-2-hexenal and 2-ethyl-1-hexanol, were repellent to *Cu. chinensis.* One might assume that this response could mean that these compounds were defensive volatiles emitted by the plant tissues, e.g., during experimental manipulation. However, the literature records do not delineate a clear pattern in that direction. For example, the analogue (E)-2-hexenal also generated a strong EAG response in both adult sexes of the chestnut weevil *C. sayi*, but did not elicit any attraction nor repellence when compared to controls [25]. The same compound generated a similar response in the strawberry leaf beetle *Galerucella vittaticollis* [26]. Another identified volatile, 2-ethyl-1-hexanol, reportedly induced an EAG response and was attractive to the four-spotted beetle *Popillia quadriguttata* [27]. Therefore, the same volatile compound might induce different behavioral responses against different insects, which is likely linked to specific adaptive habits.

Eugenol triggered no behavioral response in *Cu. chinensis*, in spite of a strong EAD response. Nonetheless, it is known that insect EAG responses to olfactory cues are not always consistent with behavior [28]. For example, the bug *Apolygus lucorum* exhibits a strong EAG response to (Z)-3-hexenyl acetate, but seemed to not be attracted to that compound in behavioral tests [29]. In order to better understand the effects of plant volatiles, behavioral studies of *Cu. chinensis* should converge the results from different approaches, both from laboratory and field observations. For example, the present study could not detect differential, sex-related behavioral responses in *Cu. chinensis* adults to each volatile, which would be consistent with a similar EAG response. Still, it is reasonable to assume that the compounds could have a different effect on each different sex, as their intrinsic biology is bound to be different. The dual attractiveness/repellency bioassay may

not be the most appropriate for such matters, and perhaps not even for the bioactivity of eugenol.

This study was the first to identify potential host plant attractants for practical applications for *Cu. chinensis* attacking *Ca. oleifera*. Monitoring and specific traps are two immediate applications warranting further development. For example, future studies could quantify the responses of *Cu. chinensis* to various chemical blends using different dosages of each of the described compounds in combination with some trap designs in order to develop an effective monitoring strategy for this pest.

**Supplementary Materials:** The following supporting information can be downloaded at: https://www.mdpi.com/article/10.3390/d14110951/s1, Figure S1: volatile total chromatograms of leaf (a), fruit (b), and seed (c) extracts, and a heatmap of the relative abundances of each identified compound (d). In the legend, F, L, and S (d) represent the fruit peels, leaves, and seeds, respectively. Table S1: Source information of selected standard compounds.

**Author Contributions:** Conceptualization, H.Q. and J.X.; methodology, S.L., E.G.P.F. and C.Q.; investigation and data curation, H.Q., D.Z. and S.L.; writing—original draft preparation, H.Q.; writing—review and editing, E.G.P.F., C.Q. and J.X. All authors have read and agreed to the published version of the manuscript.

**Funding:** This research was financially supported by the Natural Science Foundation of Guangdong province (2018A030310690) and FAPEG/CNPq (317847/2021-0).

**Institutional Review Board Statement:** Not applicable.

**Data Availability Statement:** Not applicable.

**Acknowledgments:** The authors thank Lian Tao and Lu Jian Kang for the collection of insects.

**Conflicts of Interest:** The authors alone are responsible for the content and writing of the paper. The authors declare no conflict of interest.

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
