# Peer review of "Chemical Cues Used by the Weevil Curculio chinensis in Attacking the Host Oil Plant Camellia oleifera"

_diversity, doi:10.3390/d14110951_

Round 1

Reviewer 1 Report

Chemical communication of genus Curculio is a poorly studied field however there are many hosts specific species in this genus with significance as pest insect. As the manuscript is also stated, it is important to unravel these processes, because in addition to help us understand the relationships among organisms, it can help develop methods that make sustainable agriculture feasible (especially for major pests as the one discussed in the MS).

Subject of the manuscript fits into the aims of the special issue, as its goal is to uncover host plant volatiles that mediate a very well-defined system between a host specific beetles and its host. The study presents the first steps of this process, by identifying in vitro a couple of substances that influence the behavior of the target species. However, no real ecological importance of the identified components could be concluded, so further studies are necessary.

The manuscript has the appropriate structure, the introduction describes the basic knowledge well and put main questions in a good context, however the applied methods, their description and the presentation of the results needs to be completed and somewhere explained. A minor revision therefore is definitely necessary, after which the manuscript can be accept for publication.

Reviewer 2 Report

In this study the authors attempted to investigate chemical cues involved in host plant Camellia oleifera location and recognition by the weevil Curculio chinensis by analysing the chemical composition of plant, fruit, and seed extracts from the host plant species, as well as by evaluating the antennal responses to compounds in these extracts using GC-EAD screening. Behavioural experiments under laboratory conditions were also performed. These outcomes could represent a significant contribution to the field of interest.

In my opinion the experimental protocol for the study is old fashioned. The standard procedure to collect volatile compounds to be used in electrophysiological screening is to perform headspace collection from intact plants or fruits (or with minor mechanical damage, as stressed plants volatile emissions differ from healthy ones) to ensure that only those compounds that are actually released from plants and fruits are sampled. In contrast, the authors instead extracted plant, fruit and seed samples in solvent, and the amounts of individual compounds released by the plant (fruit and seeds) cannot be estimated.

Also, authors hypothesized that chemicals from seeds serve as short-distance cues and provide key information about fruit suitability for feeding and oviposition (Line 228-231). In that case the authors must prove experimentally that compounds emitted by seeds diffuse through peel and pulp of fruit and reach insect outside. Otherwise, the experimental part with compounds released by seeds has no sense.

The condition that the insect must stay in the arm of Y-olfactometer for more than 0.5 min for its choice to be considered positive is not reliable and enough. The most reliable results are obtained when the insect reaches the odor source (or at least an arm zone beyond 2/3 of its length). If the insect has chosen the arm but does not move towards the odor source, the stimuli cannot be considered as attractive. Such insects must be excluded from the results or classified as non-responsive.

The discussion is far too long with a lot of repetition from the result section. More should be based on the peculiarities of the behaviour of the weevils. The list of references cited is poor.

My additional comments are given below:

Introduction

Provide a more detailed description of the biology of C. chinensis.b

Line 45: A single C. chinensis female can lay an average of 150 eggs. – Indicate the reference.

Line 52-54: You can’t make such suggestion on the basis of a single case.

Figure 1, I didn’t find any published data that C. chinensis release aggregation or sex pheromone. Remove the figure or add the reference.

Material and methods

Line 76: Specify the ripening stage of fruits (and seeds) used for extractions.

Line 91: The manufacturers and purity of the standards are not given.

Line100-102: What solution was used for glass capillary filling? What concentrations of extracts were used for GC_EAD screening? Please indicate the rate of constant airflow that passed through insect antenna. 5 female and 5 male antenna replications were done? Or only 5 replications for each sample concentration regardless of the sex of the insect?

Line 109: The amounts of standard compound solutions and hexane applied on filter paper are not given.

Line 119: How often did you change the stimuli in the olfactometer? Have you tested how the evaporation of the compounds used depends on time and airflow rate? Because some compounds were volatile and if you worked between 8:00 AM and 12:00 PM, the concentration of the compound in the olfactometer changed.

Was a control test performed before the choice tests? When only a stream of clean air was used during it. The control test determines whether the right and left branches are chosen equally often by insects, i. whether light and other unrecorded stimuli are properly balanced.

Results

There is some misunderstanding with two GC_EAD active compounds: trans-2-Hexen-1-ol and 2-ethyl-1-hexanol. It seems that the experiment was not done properly.

Line 175-180. Trans-2-Hexen-1-ol elicited EAG reaction in C. chinensis when fruit peel and seed extract solutions were provided. It’s odd why this compound was not active when leaves extract was provided as relative proportion of the compound in leaves extract was even higher than in seed extract.

Very similar situation with 2-ethyl-1-hexanol. This compound is present in all extracts but was GC_EAD active only when seed extract was tested, though its relative proportion in the seed extract was the smallest one.

Figure 4 Please add x axis with scale and numbers. In this figure different doses are shown, however in the legend different concentrations are mentioned.

  • the Authors excluded from the analysis the adults, which did not make a choice, but it is important to know this data. If you test e.g. 50 adults and 25 are not making a choice… the experimental set-up should be modified. It is very important question! Even if the ratio is similar to the variant of adults behaviour in an empty olfactometer.

Discussion

Line 239-246 The obtained results are compared with unpublished data that have not been reviewed. I highly recommend deleting this part of the discussion.

Round 2

Reviewer 2 Report

The manuscript had been improved and benefited; however, some questions still need to be explained. I didn’t get the answer to the methodological part of the olfactometrical bioassay. As I mentioned in the previous comments, the condition that the insect must stay in the arm of Y-olfactometer for more than 0.5 min for its choice to be considered positive (Line 140-141) is not reliable and enough. The most reliable results are obtained when the insect reaches the odor source (or at least an arm zone beyond 2/3 of its length). If the insect has chosen the arm but does not move towards the odor source, the stimuli cannot be considered as attractive. Such insects must be excluded from the results or classified as non-responsive. Insects can often switch the Y arms after 0.5 minute or later and for this reason the insect must reach the odor source not just choose the arm. Are you able to provide data how many insects chose the arm, but didn’t reach an arm zone beyond 2/3 of its length in each test?

Line 59 The cited references provide information about Diptera, Lepidoptera species not about weevils.

Lines 114-116 “Six antennae from different individuals were tested as replicates for each tested sample concentration”. It would be more informative to mark that both sexes were tested, for example ”six antennae from different individuals (males and females) were tested...” 

Lines120-121 and 134-135 duplicate each other. 

Line126   Please indicate in the manuscript how often did you change the stimuli in the olfactometer?  Your answer was “after 5 min of each test”, you mean that after every insect choice you refreshed the odor source, as some insects were observed at least for 5 min?

Response: We change the stimulus source after 5 min of each test. As we did behavioral bioassays in a room maintained at 25 ± 1℃ and the relative humidity 65 ± 5%, we did not test how the evaporation rate of used compounds depends on time and airflow rate.

Line 248 the reference is not relevant to the sentence.

In fact, there is a mess with citation in the discussion part, the references are not relevant to the discussed material.

Some different references (9, 10 and 15,16) have a uniform numbering.
